# Prospective Clinical Evaluation of the Diagnostic Accuracy of a Highly Sensitive Rapid Antigen Test Using Silver Amplification Technology for Emerging SARS-CoV-2 Variants

**DOI:** 10.3390/biomedicines10112801

**Published:** 2022-11-03

**Authors:** Kazuaki Obata, Kei Miyakawa, Toshiki Takei, Atsuhiko Wada, Yasuyoshi Hatayama, Hideaki Kato, Yayoi Kimura, Hisakuni Sekino, Junichi Katada, Akihide Ryo

**Affiliations:** 1Sekino Hospital, Tokyo 171-0014, Japan; 2Department of Microbiology, Yokohama City University School of Medicine, Kanagawa 236-0004, Japan; 3Medical Systems Research & Development Center, FUJIFILM Corporation, Kanagawa 258-8538, Japan; 4Infection Prevention and Control Department, Yokohama City University Hospital, Kanagawa 260-0004, Japan; 5Advanced Medical Research Center, Yokohama City University, Kanagawa 236-0004, Japan

**Keywords:** SARS-CoV-2, COVID-19, rapid antigen test, diagnostic performance study

## Abstract

The COVID-19 pandemic caused by SARS-CoV-2 remains a serious health concern worldwide due to outbreaks of SARS-CoV-2 variants that can escape vaccine-acquired immunity and infect and transmit more efficiently. Therefore, an appropriate testing method for COVID-19 is essential for effective infection control and the prevention of local outbreaks. Compared to reverse-transcription polymerase chain reaction (RT-PCR) tests, antigen tests are used for simple point-of-care testing, enabling the identification of viral infections. In this study, we tested the clinical usefulness of the FUJIFILM COVID-19 Ag test, an antigen test based on silver amplification and immunochromatographic technology. The FUJIFILM COVID-19 Ag test was shown to detect a lower viral concentration as compared to other conventional kits without significant performance loss in detecting prevalent SARS-CoV-2 variants. We tested nasopharyngeal and nasal swabs from a single patient during two different epidemic periods dominated by various SARS-CoV-2 variants. We observed that the sensitivity of the FUJIFILM COVID-19 Ag test was 95.7% and 85.7% in nasopharyngeal and nasal swabs, respectively. These results suggest that the FUJIFILM COVID-19 Ag test is highly sensitive and applicable when RT-PCR testing is unavailable. Furthermore, these results indicate that high-frequency testing using nasal swab specimens may be a valuable screening strategy.

## 1. Introduction

The coronavirus disease 2019 (COVID-19) pandemic caused by the severe acute respiratory syndrome coronavirus 2 (SARS-CoV-2) remains a serious health concern despite progress in the development and administration of vaccines and treatments [1,2]. Furthermore, outbreaks of SARS-CoV-2 variants with significantly altered infectivity, transmissibility, and antigenicity have been reported worldwide and have been designated as variants of concern (VOC) by the World Health Organization (WHO) [3,4]. The emergence of these variants could rapidly increase the pressure on clinical sites due to their potential to escape vaccine-induced immunity and their increased transmissibility. In addition to infection control measures (e.g., vaccinations and masks), testing symptomatic individuals with suspected COVID-19 infections and screening asymptomatic populations are essential to prevent cluster outbreaks [5,6,7]. Nucleic acid amplification tests (NATs), such as real-time RT-PCR, the gold standard for COVID-19 diagnosis, specifically detect SARS-CoV-2 RNA and have extremely high sensitivity and specificity [8]. In contrast, NATs require precise manipulation for sample preparation as well as specialized equipment and, moreover, are time-consuming to deploy. Antigen tests determine the presence of SARS-CoV-2 antigens using antibodies. Unlike NATs, this method does not require special equipment, is simple to operate, and the results are returned within 30 min. However, conventional antigen tests have lower sensitivity than NATs and a higher false negative rate [9].

The silver-enhanced immunochromatography method is unique as it combines the silver amplification process using photographic development with immunochromatography, thereby achieving high sensitivity [10,11]. The FUJIFILM COVID-19 Ag test uses silver-enhanced immunochromatography for SARS-CoV-2 diagnosis. In previous studies, we tested frozen nasopharyngeal specimens and demonstrated that the prototype of the FUJIFILM COVID-19 Ag test specifically detected SARS-CoV-2 without cross-detectivity of other human coronavirus respiratory pathogens, with a higher sensitivity than other commercially available tests [12]. However, there is no precise evaluation of the diagnostic performance against VOCs such as the Omicron strain in a prospective clinical study.

Although nasopharyngeal swab (NPS) specimens are commonly used for COVID-19 diagnosis, nasal swab (NS) specimens collected from the nostrils are increasingly used. Their collection is easier and less invasive; therefore, they are suitable for screening large populations and frequently testing individuals [13]. Reports comparing the diagnostic accuracy of antigen tests using nasopharyngeal and nasal specimens from the same patient are limited. Thus, further investigation is required to determine which samples are appropriate for antigen testing.

In this study, we investigated the sensitivity and clinical usefulness of the FUJIFILM COVID-19 Ag test to diagnose epidemic SARS-CoV-2 variants. We further compared the results from nasopharyngeal and nasal specimens from the same patients.

## 2. Materials and Methods

### 2.1. Preparation of Viruses

The SARS-CoV-2 strains listed in Table 1 were obtained from the National Institute of Infectious Diseases (Japan) and propagated in Vero-E6 cells expressing TMPRSS2 (JCRB1819), as described previously [14]. SARS-CoV-2 was quantified using RT-PCR with the NIID-N2 primer and probe set [15]. Virus samples were inactivated by adding NP-40 to a final concentration of 0.2% (*v*/*v*) prior to each immunoassay.

### 2.2. Rapid Antigen Testing of the Isolated Virus

The FUJIFILM COVID-19 Ag test was used according to the manufacturer’s instructions (Figure 1). Briefly, the samples were diluted with an extraction buffer, and one drop of the sample was added to the sample well of the FUJIFILM COVID-19 Ag test. Button 2 was immediately pressed to release a reducing reagent for the silver amplification. After the color indicator turned orange (after approximately 10–15 min), button 3 was pressed to release a silver-ion reagent to activate the silver amplification reaction. For performance comparison, four commercially available conventional rapid antigen detection kits for SARS-CoV-2 (conventional tests A–D) approved by the Ministry of Health, Labor and Welfare in Japan were used according to the manufacturer’s instructions for each detection kit.

### 2.3. Detection Limits of Antigen Rapid Diagnostic Tests (Ag-RDTs)

The detection limits of nine isolated SARS-CoV-2 strains with known viral copy numbers were evaluated by comparing the antigen tests of the FUJIFILM COVID-19 Ag test and four conventional kits. Inactivated virus samples were serially diluted two-fold with the extraction buffer supplied with each kit. Each kit was used as described in Section 2.2. The detection limits were set as the minimum number of viral copies (copies/mL) at which all would be positive, with a minimum of two independent tests at each concentration, unless the sample volume was insufficient. The test results were interpreted by at least two independent observers.

### 2.4. Prospective Clinical Evaluation

This study was conducted at the Sekino Hospital in Tokyo, Japan, over two time periods. Samples were collected between January and May 2021 and between January and March 2022. One NPS and one NS specimen were collected per patient for rapid antigen testing and one NS specimen for RT-PCR testing. We included patients who either visited the fever outpatient clinic and/or had contact with infected individuals and were referred by the local health department. Informed consent was obtained from all participants. This study was approved by the Ethics Review Committee of the Sekino Clinical Pharmacology Clinic, to which Sekino Hospital belongs (research project name: Clinical performance evaluation of rapid diagnostic reagents for novel coronaviruses, approval date: 10 December 2020).

### 2.5. Specimen Collection for Prospective Clinical Evaluation

Three specimens were collected simultaneously from the same patient by the same physician, i.e., one NPS and one NS specimen each for antigen testing and one NPS specimen for PCR testing. All specimens were collected from one nostril using the swab (Heiwamedic, Takayama-shi, Japan) included in the test kit according to the manufacturer’s instructions, and each specimen was used randomly for antigen and PCR testing. Immediately after specimen collection, antigen testing was performed according to the manufacturer’s instructions, and several individuals determined the results. NPS specimens collected for RT-PCR testing were suspended in 3 mL of saline. NPS samples were stored in the hospital at 4 °C and transferred on the same day to the clinical laboratory of SRL Corporation, an external laboratory.

### 2.6. RT-PCR Tests

The RT-PCR tests were performed by the clinical laboratory of SRL Corporation, according to the method developed by the National Institute of Infectious Diseases, Japan [15]. Viral RNA was extracted from the specimens using the QIAsymphony DSP virus/Pathogen Mini Kit (Qiagen, Hilden, Germany). Samples were tested using the QuantiTect Probe RT-PCR Kit (Qiagen). The assay results included PCR test results (positive and negative) and Ct value data (N primer results and N2 primer results), which were included in the case report.

### 2.7. Statistical Analyses

Sensitivity, specificity, positive predictive value (PPV), and negative predictive value (NPV) were calculated from the RT-PCR and antigen test results. The Clopper–Pearson method was used to calculate 95% confidence intervals (CI).

## 3. Results

### 3.1. Detection Limits of Antigen Tests

The detection limits of the FUJIFILM COVID-19 and conventional commercially available antigen tests were determined and compared for wild-type (Wuhan-Hu-1) and eight SARS-CoV-2 variants (Table 1). The detection limits of four conventional kits ranged from 5.5 to 7.0 (log_10_ copies/mL). In contrast, the detection limit of the FUJIFILM COVID-19 Ag test ranged from 4.8 to 5.5 (log_10_ copies/mL), indicating its capability to detect a lower viral concentration compared to conventional kits tested. Although the detection limits of viral copies between SARS-CoV-2 variants differed by up to about two-fold, there was no significant performance degradation in the detection of the prevalent SARS-CoV-2 variants.

### 3.2. Prospective Clinical Evaluation

NPS and NS samples were collected from 280 participants. Among these, 70 NPS specimens tested positive for SARS-CoV-2 using RT-PCR. Table 2 summarizes the demographical data and symptoms (prevalence and onset) of all participants as well as RT-PCR results (including Ct values).

The results of the FUJIFILM COVID-19 Ag test with each specimen type and the reference RT-PCR assays with NPS specimens are compared in Table 3. For NPS specimens, the sensitivity, specificity, positive predictive value (PPV), and negative predictive value (NPV) were 95.7% (95% CI: 88.0–99.1%), 100% (95% CI: 98.3–100%), 100% (95% CI: 94.6–100%), and 98.6% (95% CI: 95.9–99.7%), respectively. For NS specimens, sensitivity, specificity, PPV, and NPV were 85.7% (95% CI: 75.3–92.9%), 100% (95% CI: 98.3–100%), 100% (95% CI: 94.0–100%), and 95.5% (95% CI: 91.8–97.8%), respectively. Three NPS and ten NS samples exhibited false-negative results to reference RT-PCR (Table 4). In addition, the test using nasal specimens sometimes showed false negative results even when the Ct value was less than 20. Owing to the small number of specimens, no correlation could be found between the false-negative cases and clinical symptoms.

### 3.3. Comparison of Sensitivity between NPS and NS Specimens

Table 5 lists the sensitivities of antigen tests of the NPS and NS specimens according to the Ct values of the NPS RT-PCR test (NIID N2 primer), and Figure 2 displays the distribution of Ct values for positive or negative specimens with the antigen test. The sensitivities of NPS specimens were 100%, 100%, 100%, and 50.0% and those of NS specimens were 97.0%, 88.0%, 83.3%, and 16.7% with Ct values of <20, 20–25, 25–30, and >30, respectively. For the NPS specimens, the sensitivity was 100% in specimens with a Ct value of ≤30, which was comparable to that of RT-PCR in the reference assay. In contrast, for NS specimens, the sensitivity was 92.2% in specimens with Ct values of ≤30, which was slightly lower than that for NPS specimens. Relationship between Ct values of RT-PCR and days after onset and their antigen rapid diagnostic tests are shown in Appendix A.

The study was conducted over two periods; 208 specimens were collected from participants during the third and fourth waves of the COVID-19 pandemic between January and May 2021, and 72 specimens were collected during the sixth wave between January and March 2022 in Japan. In the first study period, 37 samples were positive, and 6 samples were false negative; in the second period, 33 samples were positive, and 4 samples were false negative. The results (Table 6) suggested that the sensitivity at Ct values <30 did not change during the test period. In addition, the specificity was 100% in both periods, regardless of specimen type.

## 4. Discussion

Compared to RT-PCR, the gold standard for COVID-19 diagnosis, the antigen test presents advantages such as simplicity of operation and faster results (within 30 min). Since it does not require complex operations like RT-PCR, the risk of specimen contamination during testing could be reduced. While quantitative antigen tests such as fluorescence detection have similar advantages, they require the installation of expensive specialized equipment and a refrigerated storage environment since their reagents can often not be stored at room temperature for long periods of time. However, qualitative antigen tests are less sensitive than RT-PCR and quantitative antigen testing. Indeed, according to a Cochrane meta-analysis, during the first year of the COVID-19 pandemic, the average sensitivity and specificity of antigen tests were 56.2% and 99.5%, respectively [16]. In addition, the latest meta-analysis results indicate that higher performance has been achieved due to the improved quality of the qualitative antigen tests [17]. Nevertheless, the average sensitivity is 72.0%, and further technical improvement is required. The FUJIFILM COVID-19 Ag test used in this study employs a silver-enhanced immunochromatography method that applies the process of silver amplification by photo-development and thus is expected to be more sensitive than other conventional methods of antigen testing.

To systematically evaluate the sensitivity and clinical usefulness of the FUJIFILM COVID-19 Ag test against epidemic VOC, we evaluated specimens of 280 individuals. These individuals either visited the fever outpatient clinic for suspected SARS-CoV-2 infection or were certified by the health center as being in close contact with a patient with COVID-19 and recommended for testing. We observed a high detection sensitivity of 95.7% and 85.7% for NPS and NS specimens, respectively, in the 70 participants that were positive as detected by RT-PCR. The specificity of nasopharyngeal and nasal specimens was 100% for the 210 participants who tested negative by RT-PCR, and no false positive results were detected. These results indicate that the FUJIFILM COVID-19 Ag test is more sensitive than other antigen testing methods used in the Cochrane meta-analysis [16,17]. Notably, the sensitivity was 100% for nasopharyngeal specimens with a Ct value ≤30, indicating that the detection sensitivity was close to that of RT-PCR. In contrast, for Ct values >30, the sensitivity was 50%, and the overall sensitivity was lower than that of RT-PCR. In many cases, replicative SARS-CoV-2 cannot be isolated from specimens with Ct values above 30, suggesting that the threshold for infectivity and transmissibility of SARS-CoV-2 corresponds with the Ct value of 30 [18,19]. The prompt identification of carriers of viable SARS-CoV-2 and the prevention of secondary infections and clusters is the most important step in controlling a local epidemic [20]. Therefore, detecting cases of viral infection corresponding with a Ct value ≤30 is important. Above all, rapid and easy-to-use antigen tests are valuable in identifying individuals who may be carriers of the infection. Our current results demonstrate that the FUJIFILM COVID-19 Ag test has sufficient performance to be utilized for surveillance and outbreak prevention.

Two specimen types, NPS and NS, are often used for antigen testing. Our results demonstrated that the sensitivity of antigen testing with NS specimens to the reference RT-PCR method with NPS specimens was observed to be lower than that with NPS specimens. Since the nasal cavity generally contains fewer viruses than the nasopharynx [21], the sensitivity of tests using NS specimens may be lower, particularly in cases with relatively low viral load. In addition, our current result shows that the antigen testing with nasal specimens exhibited a false negative even when the Ct value was less than 20 in the RT-PCR testing with nasopharyngeal specimens. Although NPS and NS specimens were collected at the same time by the same physician to reduce the variability in the specimen collection, some cases of poor correlation between viral loads in NS and NPS specimens have been reported, and this false-negative case in this study may have occurred in participants with low viral loads in NS specimens [21]. Thus, antigen tests conducted using NS specimens are less sensitive than those conducted using NPS specimens. However, testing with NS specimens has the advantage of being simpler and less invasive because it allows self-collection of specimens and reduces the risk of infection for healthcare workers. Simulation studies suggest that frequency and specificity, in addition to detection sensitivity, are essential for a screening test [22]. Therefore, testing with NS specimens may be suitable for screening tests because it allows for increased frequency. Further, careful research is needed to explore this hypothesis.

In this study, the results of NS specimens with the FUJIFILM COVID-19 Ag test exhibited an overall sensitivity of >90% for samples with Ct values <30, which is more sensitive than other antigen tests [21,23], and a specificity of 100%. Simulation studies suggest that highly sensitive antigen tests can reduce the number of tests required, shorten the isolation period of patients, and accelerate their return to society [24]. Collectively, these results suggest that the FUJIFILM COVID-19 Ag test using NPS specimens had a similar sensitivity to RT-PCR in detecting highly infectious individuals with a Ct value of ≤30, while using NS specimens may be a useful screening method because it is less invasive and can be easily performed at high frequencies.

Since the COVID-19 pandemic, SARS-CoV-2 has acquired several mutations, resulting in the emergence of variants [1]. In this study, we evaluated the detection limit of isolated SARS-CoV-2 strains identified as VOC by the WHO [3]. Although the detection limit was revealed to vary by up to two-fold depending on the variant species, the FUJIFILM COVID-19 Ag test was able to detect infections with a lower viral load than the conventional antigen test, which could not. We conducted the clinical study from January to May 2021, when the Wuhan and Alpha strains were prevalent, and from January to March 2022, when the Omicron BA.1 and BA.2 strains were endemic [25,26]. We observed no significant difference in sensitivity and specificity between the two study periods. This suggests that the FUJIFILM COVID-19 Ag test results are not dependent on the SARS-CoV-2 variant. Frequent mutations have been identified in the gene encoding the spike protein, which has led to increased infectivity and the ability to evade natural infection and vaccine-neutralizing antibody responses [27]. In comparison, fewer mutations were observed in the gene encoding the nucleocapsid protein [28]. Most antigen tests, including this FUJIFILM COVID-19 Ag test, target the nucleocapsid protein for detection. Therefore, a decrease in reactivity across variants is unlikely [12]. These results suggest that the FUJIFILM COVID-19 Ag test will be valuable in diagnosing prevalent SARS-CoV-2 variants.

## Figures and Tables

**Figure 1 biomedicines-10-02801-f001:**
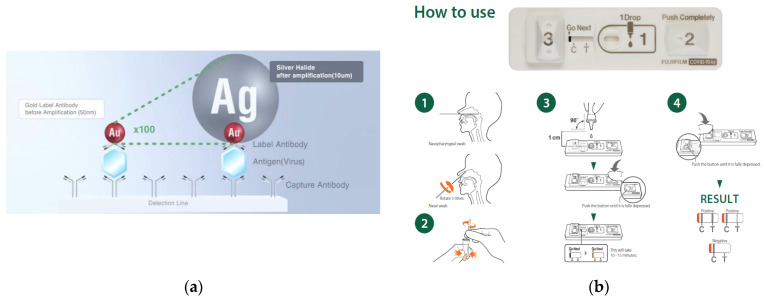
Details of the FUJIFILM COVID-19 Ag test. (**a**) The principle of silver amplification technology: colloidal gold particles are amplified approximately 100 times, increasing the detection sensitivity; (**b**) Instructions for using the FUJIFILM COVID-19 Ag test. Collected specimens were diluted with an extraction buffer in the tube included in the package, and one drop of the sample was added onto the sample well of the FUJIFILM COVID-19 Ag test. Button 2 was pressed immediately to release reducing reagents for the silver amplification. After the Go Next indicator mark turned orange (after about 10–15 min), button 3 was pressed to release Ag ions to activate the silver amplification reaction.

**Figure 2 biomedicines-10-02801-f002:**
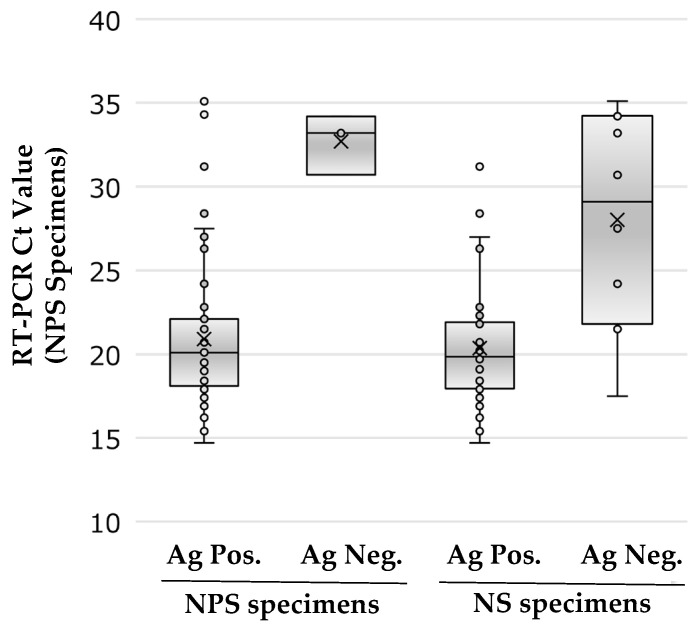
Antigen testing and RT-PCR results of all RT-PCR positive subjects. All antigen testing results for NPS and NS specimens are shown as Ct-value on the *y*-axis (left side: positive antigen testing (Ag Pos.) results; right side: negative antigen testing (Ag Neg.) results) and target specimens based on the antigen testing result on the *x*-axis. Each plot shows the results for each subject.

**Table 1 biomedicines-10-02801-t001:** Detection limits of the FUJIFILM COVID-19 Ag and conventional tests of isolated SARS-CoV-2 variants.

SARS-CoV-2 Strain Name	WHO Label/ Pango Lineage	Detection Limit (log_10_ Copies/mL)				
		FUJIFILM COVID-19 Ag Test	Conventional Test A	Conventional Test B	Conventional Test C	Conventional Test D
2019-nCoV/Japan/TY/WK-521/2020	Wild type	5.5	7.0	6.7	5.8	6.7
hCoV-19/Japan/QHN001/2020	Alpha/B.1.1.7	5.1	6.0	6.3	5.7	N.T.
hCoV-19/Japan/TY8-612/2021	Beta/B.1.351	5.4	6.3	6.6	6.0	N.T.
hCoV-19/Japan/TY7-503/2021	Gamma/P.1	4.8	6.0	6.0	5.7	N.T.
hCoV-19/Japan/TY11-927/2021	Delta/B.1.617.2	5.2	6.7	6.4	5.5	6.4
hCoV-19/Japan/TY38-873/2021	Omicron/BA.1	5.4	6.6	6.9	5.7	6.0
hCoV-19/Japan/TY40-385/2022	Omicron/BA.2	5.5	6.7	N.T.	5.8	6.7
hCoV-19/Japan/TY41-686/2022	Omicron/XE	5.5	6.4	N.T.	5.8	6.4
hCoV-19/Japan/TY41-702/2022	Omicron/BA.5	5.2	6.4	N.T.	N.T.	6.4

**Table 2 biomedicines-10-02801-t002:** Summary of demographical data and symptoms of the study population.

	Total	RT-PCR Positive	RT-PCR Negative
N	280	70	210
Age (years)	40.1 ± 14.7	38.8 ± 13.6	40.5 ± 15.1
Male	143 (51%)	43 (61%)	100 (48%)
Days from the onset	2.2 ± 2.1 (n = 127)	2.6 ± 2.4 (n = 52)	1.9 ± 1.7 (n = 75)
Ct value (NIID N1)	-	25.5 ± 4.5	-
Ct value (NIID N2)	-	21.4 ± 4.7	-
Cough	84 (30.0%)	40 (57.1%)	44 (21.0%)
Sore throat	90 (32.1%)	39 (55.7%)	51 (24.3%)
Headache	112 (40.0%)	42 (60.0%)	70 (33.3%)
Runny nose	18 (6.4%)	4 (5.7%)	14 (6.7%)
Sputum production	48 (17.1%)	24 (34.3%)	24 (11.4%)
Parosmia	13 (4.6%)	5 (7.1%)	8 (3.8%)
Dysgeusia	25 (8.9%)	9 (12.9%)	16 (7.6%)
Arthralgia	15 (5.4%)	8 (11.4%)	7 (3.3%)

**Table 3 biomedicines-10-02801-t003:** Results of the FUJIFILM COVID-19 Ag test on nasopharyngeal swab (NPS) and nasal swab (NS) specimens.

**NPS Specimens**		**NPS RT-PCR (NIID N2)**	
		**Positive**	**Negative**
FUJIFILM COVID-19 Ag test	Positive	67	0
	Negative	3	210
Sensitivity (%)		95.7 (88.0–99.1)	
Specificity (%)		100 (98.3–100)	
Positive Predictive Value (%)		100 (94.6–100)	
Negative Predictive Value (%)		98.6 (95.9–99.7)	
**NS Specimens**		**NPS RT-PCR (NIID N2)**	
		**Positive**	**Negative**
FUJIFILM COVID-19 Ag test	Positive	60	0
	Negative	10	210
Sensitivity (%)		85.7 (75.3–92.9)	
Specificity (%)		100 (98.3–100)	
Positive Predictive Value (%)		100 (94.0–100)	
Negative Predictive Value (%)		95.5 (91.8–97.8)	

**Table 4 biomedicines-10-02801-t004:** Detailed data of 10 cases with false-negative results to reference RT-PCR.

Case	Date of Specimen Collection	Age (Years)	Sex	Days from the Onset	Body Temperature (°C)	Symptoms	RT-PCR Ct Value		FUJIFILM COVID-19 Ag Test	
							NIID N2	NIID N1	NPS	NS
1	Feb 2021	26	M	N.D.	39.5	Sore throat, headache, diarrhea	17.5	23.1	Pos.	Neg.
2	Feb 2021	33	F	1	36.8	Sore throat	21.5	25.2	Pos.	Neg.
3	Feb 2021	32	M	11	36.6	Arthralgia, dysgeusia, parosmia	35.1	39.6	Pos.	Neg.
4	Feb 2021	33	F	8	37.6	-	34.3	N.D.	Pos.	Neg.
5	Apr 2021	31	M	N.D.	37.5	-	24.2	30.9	Pos.	Neg.
6	May 2021	30	M	6	37.0	Cough, sore throat, dysgeusia	21.9	26.7	Pos.	Neg.
7	Jan 2022	32	F	N.D.	35.4	-	30.7	34.9	Neg.	Neg.
8	Feb 2022	34	M	0	37.7	Cough, sore throat	33.2	37.2	Neg.	Neg.
9	Feb 2022	33	F	N.D.	36.6	Cough, sore throat, sputum, runny nose	34.2	37.6	Neg.	Neg.
10	Mar 2022	41	M	0	38.8	Cough, sore throat, arthralgia	27.5	31.2	Pos.	Neg.

N.D.: No Data, M: male, F: female.

**Table 5 biomedicines-10-02801-t005:** Sensitivity of the FUJIFILM COVID-19 Ag test according to the Ct value.

RT-PCR (NPS Specimens)		Sensitivity (%)	
Ct Value (NIID N2)	N	NPS Specimens	NS Specimens
<20	33	100 (89.4–100)	97.0 (84.2–99.9)
20–25	25	100 (86.3–100)	88.0 (68.8–97.5)
25–30	6	100 (54.1–100)	83.3 (35.9–99.6)
>30	6	50.0 (11.8–88.2)	16.7 (0.4–64.1)

**Table 6 biomedicines-10-02801-t006:** Sensitivity of the FUJIFILM COVID-19 Ag test according to the study period.

RT-PCR (NPS Specimens)	January 2021–May 2021			January 2022–March 2022		
	Sensitivity (%)					
Ct Value (NIID N2)	N	NPS Specimens	NS Specimens	N	NPS Specimens	NS Specimens
<20	18	100	94.4	15	100	100
20–25	12	100	75.0	13	100	100
25–30	4	100	100	2	100	50.0
>30	3	100	33.3	3	0	0

## Data Availability

Data is contained within the article and the Appendix A.

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
