# Peer review of "Prospective Clinical Evaluation of the Diagnostic Accuracy of a Highly Sensitive Rapid Antigen Test Using Silver Amplification Technology for Emerging SARS-CoV-2 Variants"

_biomedicines, 2022, doi:10.3390/biomedicines10112801_

Round 1

Reviewer 1 Report

The manuscript by Kazuaki Obata et al describes clinical validation of FUJIFILM COVID-19 Ag test. The article could be of interest for medical specialists, but should be further improved.

It is necessary to better structure the presented tables, so that the text referring to them is nearby. So, for example, on page 4 there is a reference to table 1 from page 2. In this version of the manuscript layout reading tables could be complicated.

The discussion section could be improved and better structured. It is not clear what are the advantages and limitations of the FUJIFILM COVID-19 Ag test compared to other rapid detection platforms. Table 4, which contains discordant SARS-CoV-2 detection results, is poorly discussed.

Also, the article would benefit from a more detailed comparison of this method and competitive platforms, in the terms of assay performance after long storage and ability to correctly analyze contaminated samples.

Author Response

Response to the Reviewer#1

The manuscript by Kazuaki Obata et al describes clinical validation of FUJIFILM COVID-19 Ag test. The article could be of interest for medical specialists, but should be further improved.

Response We appreciate the valuable and constructive comments provided by the reviewer. The followings are our point-by-point responses:

It is necessary to better structure the presented tables, so that the text referring to them is nearby. So, for example, on page 4 there is a reference to table 1 from page 2. In this version of the manuscript layout reading tables could be complicated.

Response We are thankful to you for pointing out the issues with the layout of the manuscript. As you suggested, we have moved Table1 from page 2 to page 4 and corrected the positions of the text and figures in the manuscript so that they do not diverge.

The discussion section could be improved and better structured. It is not clear what are the advantages and limitations of the FUJIFILM COVID-19 Ag test compared to other rapid detection platforms.

Response Thank you for your suggestions for improving this paper. We have modified the structure of the Discussion section to make it as clear as possible. Then, in addition to the PCR method, a comparison of the advantages of this test method over antigen quantification methods, a rapid diagnostic device using fluorescence detection, was added (Page 7 Line 216–220).

Table 4, which contains discordant SARS-CoV-2 detection results, is poorly discussed.

Response We apologize for the lack of discussion of this aspect of the manuscript. No new interpretations of discordant cases were generated from clinical symptoms or participant information, but a description of the results in Table 4 is included in the manuscript (Page 5, Line 170–174). We also discussed the reasons for false negative results in tests using nasal specimens, even when Ct values were below 20 (Page 7 Line 259-261).

Also, the article would benefit from a more detailed comparison of this method and competitive platforms, in the terms of assay performance after long storage and ability to correctly analyze contaminated samples.

Response We are thankful to you for your suggestion for the comparison with other competitive platforms. Compared to the competitive PCR method, the antigen test has the advantage of being able to be stored at room temperature for a long period of time, and because it does not involve complicated operations, the risk of sample contamination is reduced. We have included a description of these comparisons in the Discussion section (Page 7 Line 219–222).

Reviewer 2 Report

Thank you very much to the editor of Biomedicines for allowing me to review the document titled “Prospective clinical evaluation of the diagnostic accuracy of a highly sensitive rapid antigen test using silver amplification technology for emerging SARS-CoV-2 variants.”

The paper written by Obata and colleagues is based on the valuation of the sensitivity and clinical usefulness of the FUJIFILM COVID-19 Ag test to diagnose epidemic SARS-CoV-2 variants. Further, the authors compared the results of Ag test from nasopharyngeal and nasal specimens collected in two different periods from the 280 patients, using RT-PCR as reference.

The study is well designed with a rich and new literature, the methods and result are clear such as the tables and the figures that are easy to understand for the reader. For this reason, in my opionion, this paper is interesting and it deserves publication

I have a few questions and comments regarding the key findings reported in the manuscript:

1. In this study, a total of 280 participants were recruited between January and May 2021 and between January and March 2022. Can the authors indicate how many participants were recruited for each period?

2. Lane 232: “We observed no significant difference in sensitivity and specificity between the two study periods”. This sentence is justified by the fact that the authors noticed an equal distribution of false negatives in the two different periods? The authors could include this information in the results.

3. Lane 231: The authors observed no significant different in sensibility and specificity. As reported in the discussion of results, the detection limit of the test can range from 4.8 to 5.5 log10 copies/ml. Specifically, considering the prevalent variants in the two sampling periods, we have 5.1 for the Alpha variant and 5.5 for the Omicron variant, as reported in Tab.1. According to my interpretation of the data, the Lod for the Alpha variant is about 125 copies/µl, and about 300 copies/µl for the Omicron one. Is my interpretation correct? If yes, the LOD of the Alpha variant is about half that of the Omicron variant, which is not irrelevant and needs to be elaborated more clearly in the discussion.

4.  Lane 208-215: Please clarify the concept.

Author Response

Response to the Reviewer#2

Thank you very much to the editor of Biomedicines for allowing me to review the document titled “Prospective clinical evaluation of the diagnostic accuracy of a highly sensitive rapid antigen test using silver amplification technology for emerging SARS-CoV-2 variants.”The paper written by Obata and colleagues is based on the valuation of the sensitivity and clinical usefulness of the FUJIFILM COVID-19 Ag test to diagnose epidemic SARS-CoV-2 variants. Further, the authors compared the results of Ag test from nasopharyngeal and nasal specimens collected in two different periods from the 280 patients, using RT-PCR as reference.The study is well designed with a rich and new literature, the methods and result are clear such as the tables and the figures that are easy to understand for the reader. For this reason, in my opionion, this paper is interesting and it deserves publication. I have a few questions and comments regarding the key findings reported in the manuscript:

Response We appreciate your interest in our paper and deeply appreciate your recognition of it as worthy of publication. The answers to the questions and comments received are as follows.

  1. In this study, a total of 280 participants were recruited between January and May 2021 and between January and March 2022. Can the authors indicate how many participants were recruited for each period?

Response We apologize for the lack of clarity in the breakdown of the study participants. The number of study participants for each period was n = 208 for January–May 2021 and n = 72 for January–March 2022, which is described on Page 6, line 196-199. However, since it is difficult to understand that they are research participants in the original description, the description has been slightly revised to more clearly highlight this information.

  1. Lane 232: “We observed no significant difference in sensitivity and specificity between the two study periods”. This sentence is justified by the fact that the authors noticed an equal distribution of false negatives in the two different periods? The authors could include this information in the results.

Response We are thankful for your straightforward point regarding the validity of the content of the Discussion section. In the first study period, there were six false negative samples among the 37 positives, and in the second period, there were four false negative samples among the 33 positives. Although the sample size is too small to precisely analyze the distribution, the number of false negatives can be considered roughly equivalent.

Since there was no description of the number of false negatives, we have included this information in the Results section (Page 6, Line 185–186) according to the reviewer's comments.

  1. Lane 231: The authors observed no significant different in sensibility and specificity. As reported in the discussion of results, the detection limit of the test can range from 4.8 to 5.5 log10 copies/ml. Specifically, considering the prevalent variants in the two sampling periods, we have 5.1 for the Alpha variant and 5.5 for the Omicron variant, as reported in Tab.1. According to my interpretation of the data, the Lod for the Alpha variant is about 125 copies/µl, and about 300 copies/µl for the Omicron one. Is my interpretation correct? If yes, the LOD of the Alpha variant is about half that of the Omicron variant, which is not irrelevant and needs to be elaborated more clearly in the discussion

Response Thank you for this accurate, straight-to-the-point, and very helpful comment. Your interpretation is mostly correct, and the numbers of copies detected in the isolated and inactivated alpha and omicron variants are approximately two-fold different. On the other hand, during the first half of this clinical evaluation, the Wuhan and Alpha strains each accounted for about half of the total number of cases in Japan, and although it is not clear to what extent this is due to the fact that this is a prospective trial using fresh specimens, the clinical sensitivities and specificities for the two periods are comparable.

In response to this point, we have changed the description to state that the difference was about two-fold, since the LoD with the isolated virus was not equivalent. (Page 4, Line 146–149、Page 8, Line 284–287)

  1. Lane 208-215: Please clarify the concept.

Response We apologize for the lack of clarity regarding the concept. We think that the main reason for the low sensitivity of nasal specimens is the low levels of the virus in the specimens; therefore, we have modified the context to make the logical structure easier to understand (Page 7, Line 257–266). In addition, to clarify the difference in detection performance for different specimens, we have added Figure 2, which details the Ct value distribution of specimens that were positive and negative in the antigen test, to the text.

Reviewer 3 Report

The presented paper "Prospective clinical evaluation of the diagnostic accuracy of a highly sensitive rapid antigen test using silver amplification 3 technology for emerging SARS-CoV-2 variants" by Obata et al. is quite important in the context of the COVID-19 pandemic. Authors tested the clinical usefulness of the FUJIFILM COVID-19 Ag test, an antigen test based on silver amplification and immunochromatographic technology. The FUJIFILM COVID-19 Ag test was shown to detect a lower viral concentration as compared to other conventional kits without significant performance loss in detecting prevalent SARS-CoV-2 variants. Researchers tested nasopharyngeal and nasal swabs from a single patient during two different epidemic periods dominated by various SARS-CoV-2 variants.

Minor points

Materials and Methods

Line 100 It is not clear how many repetitions were performed at each of the tested virus concentrations in determining the Limit of detection. The description that the test results were interpreted by at least two independent observers is not enough to understand exactly how the sensitivity threshold was determined. Was it absolute or not.

Line 109 One NPS and one NS specimen were collected per patient for rapid antigen testing and one NS specimen for RT-PCR testing. It is not clear how the materials were collected. Whether they were collected from one nostril or two. How the sample was separated for PCR analysis and antigen testing? It needs to be explained.

Results generally OK

Discussion

The disadvantage of antigenic tests is their lower sensitivity compared to RT-PCR. According to a Cochrane meta-analysis, in the first year, the sensitivity of antigen tests was 56.2% (95% CI, 29.5% to 79.8%) and the mean specificity was 99.5% (95% CI, range 98.1% to 99.9%; based on eight experiments in five studies on 943 samples) [Rapid, point-of-care antigen and molecular-based tests for diagnosis of SARS-CoV-2 infection. Cochrane Database Syst. Rev. 2021, 2021, CD013705]. More recent meta-analyses show better numbers, which may indicate an improvement in the quality of the tests used and the achievement of higher analytical performance. For example, a recent meta-analysis analyzed 194 studies with a total of 221,878 tests performed [Accuracy of rapid point-of-care antigen-based diagnostics for SARS-CoV-2: An updated systematic review and meta-analysis with meta-regression analyzing influencing factors. PLOS Med. 2022 May 26;19(5):e1004011. doi: 10.1371/journal.pmed.1004011. PMID: 35617375; PMCID: PMC9187092.]. Overall, pooled sensitivity and specificity estimates showed values 72.0% (95% confidence interval [CI] 69.8 to 74.2) and 98.9% (95% CI 98.6 to 99.1) respectively. When the manufacturer's instructions were followed, sensitivity increased to 76.3% (95% CI 73.7-78.7). The results obtained should be discussed in the context of available meta-analyses. In addition, there is no discussion, important for antigenic tests, about the usefulness of tests specifically for identifying carriers of an infectious virus. The Value of Rapid Antigen Tests for Identifying Carriers of Viable SARS-CoV-2. Viruses 2021, 13, 2012. https://doi.org/10.3390/v13102012. These links should be added to the discussion and discussed in the context of the results obtained.

Author Response

Response to the Reviewer#3

The presented paper "Prospective clinical evaluation of the diagnostic accuracy of a highly sensitive rapid antigen test using silver amplification 3 technology for emerging SARS-CoV-2 variants" by Obata et al. is quite important in the context of the COVID-19 pandemic. Authors tested the clinical usefulness of the FUJIFILM COVID-19 Ag test, an antigen test based on silver amplification and immunochromatographic technology. The FUJIFILM COVID-19 Ag test was shown to detect a lower viral concentration as compared to other conventional kits without significant performance loss in detecting prevalent SARS-CoV-2 variants. Researchers tested nasopharyngeal and nasal swabs from a single patient during two different epidemic periods dominated by various SARS-CoV-2 variants.

Response We appreciate your understanding of our paper. The answers to the minor points you have raised are as follows:

Minor points

Materials and Methods

Line 100 It is not clear how many repetitions were performed at each of the tested virus concentrations in determining the Limit of detection. The description that the test results were interpreted by at least two independent observers is not enough to understand exactly how the sensitivity threshold was determined. Was it absolute or not.

Response We apologize for not providing a detailed description of the protocol for determining the lower limit of detection. Thank you for your straightforward advice regarding this. To determine the detection limit, unless there were insufficient samples, we conducted at least two tests at each concentration and set the concentration at which all the tests are positive as the detection limit concentration. We have included a description of this information in the Materials and Methods section (Page 3 Line 101–103).

Line 109 One NPS and one NS specimen were collected per patient for rapid antigen testing and one NS specimen for RT-PCR testing. It is not clear how the materials were collected. Whether they were collected from one nostril or two. How the sample was separated for PCR analysis and antigen testing? It needs to be explained.

Response We apologize for not clearly describing the method of specimen collection. A total of three specimens (NPS and NS specimen for rapid antigen test and NS specimen for RT-PCR test) were all collected from one nostril of each participant. The two NPS samples collected were used randomly without any distinction between PCR and antigen testing. We have included a clearer description of the collection method used in the manuscript (Page 3, Line 117–121).

Results generally OK

Discussion

The disadvantage of antigenic tests is their lower sensitivity compared to RT-PCR. According to a Cochrane meta-analysis, in the first year, the sensitivity of antigen tests was 56.2% (95% CI, 29.5% to 79.8%) and the mean specificity was 99.5% (95% CI, range 98.1% to 99.9%; based on eight experiments in five studies on 943 samples) [Rapid, point-of-care antigen and molecular-based tests for diagnosis of SARS-CoV-2 infection. Cochrane Database Syst. Rev. 2021, 2021, CD013705]. More recent meta-analyses show better numbers, which may indicate an improvement in the quality of the tests used and the achievement of higher analytical performance. For example, a recent meta-analysis analyzed 194 studies with a total of 221,878 tests performed [Accuracy of rapid point-of-care antigen-based diagnostics for SARS-CoV-2: An updated systematic review and meta-analysis with meta-regression analyzing influencing factors. PLOS Med. 2022 May 26;19(5):e1004011. doi: 10.1371/journal.pmed.1004011. PMID: 35617375; PMCID: PMC9187092.]. Overall, pooled sensitivity and specificity estimates showed values 72.0% (95% confidence interval [CI] 69.8 to 74.2) and 98.9% (95% CI 98.6 to 99.1) respectively. When the manufacturer's instructions were followed, sensitivity increased to 76.3% (95% CI 73.7-78.7). The results obtained should be discussed in the context of available meta-analyses. In addition, there is no discussion, important for antigenic tests, about the usefulness of tests specifically for identifying carriers of an infectious virus. The Value of Rapid Antigen Tests for Identifying Carriers of Viable SARS-CoV-2. Viruses 2021, 13, 2012. https://doi.org/10.3390/v13102012. These links should be added to the discussion and discussed in the context of the results obtained.

Response We are thankful for this suggestion. As the reviewer has pointed out, The literature presented was cited to compare and discuss the results of the meta-analysis analysis with the results of this study (Page 7, Line 223–231、Page 8, Line 240–241). In addition to that, we have added appropriate citations to the literature you mentioned in our discussion of the usefulness of tests to identify carriers of infectious viruses (Page 8, Line 247–249).